# Anti-Proliferative Properties of the Novel Hybrid Drug Met-ITC, Composed of the Native Drug Metformin with the Addition of an Isothiocyanate H_2_S Donor Moiety, in Different Cancer Cell Lines

**DOI:** 10.3390/ijms242216131

**Published:** 2023-11-09

**Authors:** Valentina Citi, Elisabetta Barresi, Eugenia Piragine, Jacopo Spezzini, Lara Testai, Federico Da Settimo, Alma Martelli, Sabrina Taliani, Vincenzo Calderone

**Affiliations:** 1Department of Pharmacy, University of Pisa, Via Bonanno 6, 56126 Pisa, Italy; valentina.citi@unipi.it (V.C.); elisabetta.barresi@unipi.it (E.B.); eugenia.piragine@unipi.it (E.P.); jacopo.spezzini@phd.unipi.it (J.S.); lara.testai@unipi.it (L.T.); federico.dasettimo@unipi.it (F.D.S.); sabrina.taliani@unipi.it (S.T.); vincenzo.calderone@unipi.it (V.C.); 2Center for Instrument Sharing of the University of Pisa (CISUP), University of Pisa, Lungarno Pacinotti 43/44, 56126 Pisa, Italy; 3Interdepartmental Research Center “Biology and Pathology of Ageing”, University of Pisa, 56126 Pisa, Italy

**Keywords:** metformin, hydrogen sulfide, hybrid drugs, H_2_S donor, isothiocyanate

## Abstract

Metformin (Met) is the first-line therapy in type 2 diabetes mellitus but, in last few years, it has also been evaluated as anti-cancer agent. Several pathways, such as AMPK or PI3K/Akt/mTOR, are likely to be involved in the anti-cancer Met activity. In addition, hydrogen sulfide (H_2_S) and H_2_S donors have been described as anti-cancer agents affecting cell-cycle and inducing apoptosis. Among H_2_S donors, isothiocyanates are endowed with a further anti-cancer mechanism: the inhibition of the histone deacetylase enzymes. On this basis, a hybrid molecule (Met-ITC) obtained through the addition of an isothiocyanate moiety to the Met molecule was designed and its ability to release Met has been demonstrated. Met-ITC exhibited more efficacy and potency than Met in inhibiting cancer cells (AsPC-1, MIA PaCa-2, MCF-7) viability and it was less effective on non-tumorigenic cells (MCF 10-A). The ability of Met-ITC to release H_2_S has been recorded both in cell-free and in cancer cells assays. Finally, its ability to affect the cell cycle and to induce both early and late apoptosis has been demonstrated on the most sensitive cell line (MCF-7). These results confirmed that Met-ITC is a new hybrid molecule endowed with potential anti-cancer properties derived both from Met and H_2_S.

## 1. Introduction

Metformin (Met) today represents the first-line therapy for the treatment of type 2 diabetes mellitus (T2DM) [1]. It can delay the progression of T2DM, decrease mortality rates in diabetic patients by reducing glucose synthesis, and improve insulin sensitivity in peripheral tissues by activating insulin receptor expression and enhancing its tyrosine kinase activity [2]. Met reduces the onset of comorbidities in patients with T2DM because it counteracts vascular oxidative stress and inflammation, preserving the functionality of endothelium [3]. Moreover, Met is emerging as a therapy able to limit body weight gain by influencing lipid metabolism [4]. The mechanism of action of Met, although it has been approved since 1994 by the FDA, is not fully clarified: until now, Met is recognized as an activator of adenosine monophosphate-activated protein kinase (AMPK), which is a kinase that influences several metabolic processes including glucose and lipid metabolism, energy homeostasis, and insulin sensitivity [5]. In the last decade, many studies and clinical trials have reported that Met reduces the incidence of cancer by approximately 10% to 40% in patients with T2DM compared to patients who did not receive Met and, when administered simultaneously to anti-cancer drugs, increases the response rate to anti-cancer drugs and reduces cancer mortality [6]. These effects could be attributed to the Met-mediated AMPK activation, which induces cell cycle G1 phase arrest [7], promotes apoptosis, and inhibits mTOR activity, reducing tumor growth and progression [8,9,10]. Furthermore, cancer cells are greedy for glucose, as it is essential for supporting their rapid and invasive proliferation; Met, which efficaciously reduces glucose blood levels, may limit tumor cell growth by slowing glycolysis, leading to a significant production of lactate, which lowers intracellular pH and leads to cell death [11]. The urgent need to develop innovative strategies for cancer management has led research to focus attention on novel potential strategies: hydrogen sulfide is emerging as a fundamental endogenous gas mediator involved in several physiopathological processes [12]. Indeed, H_2_S has been demonstrated to promote anti-cancer effects in several tumor types by blocking the cell cycle in the G2 phase, arresting cell proliferation through the promotion of apoptosis, necrosis and autophagy, and inhibiting invasion and migration processes [13]. Several synthetic molecules able to “donate” H_2_S have been described and their potential efficacy in the prevention and treatment of cancer has been reported [14,15]. Furthermore, the anti-cancer properties of several dietary compounds have been widely investigated in recent years. Among these, it has been reported that the consumption of *Brassicaceae*, to which broccoli, rocket salad, and cauliflower belong, reduces the incidence of cancer and promotes a better response to anti-cancer drugs [16]. These effects are potentially due to the high content of glucosinolates, which are converted into isothiocyanates (ITCs) by the activity of the enzyme myrosinase, present both in *Brassicaceae* cells and in the mammalian gut [17]. ITCs, both natural and synthetic, are reported to be H_2_S donors and promote anti-cancer activities by sharing with H_2_S the same mechanisms of action [18,19,20]. Generally, moieties able to donate gasotrasmitters can be pharmacologically exploited alone or can be used to develop hybrid drugs, i.e., new chemical entities that evoke the pharmacological effect of the “native” drug in addition to the beneficial effects of the gasotransmitter [21,22]. This strategy was born with nitric oxide, leading to highly successful new pharmacological tools for the treatment of cardiovascular pathologies [23,24,25,26,27,28] and continued with H_2_S. Several H_2_S donor moieties have been used for the development of hybrid H_2_S donor compounds useful for various diseases [29,30,31,32,33,34,35,36,37], including cancer [38,39,40]. Based on this evidence, the ITC moiety represents a very attractive chemical entity to be exploited for this purpose. Thus, in this work, a completely new hybrid H_2_S donor—metformin (Met-ITC)—was synthetized featuring the ITC moiety and, as a first step, the characterization of the H_2_S donor properties was performed. Moreover, the anti-proliferative properties on pancreatic and breast cancer cell lines were investigated, exploring the possible mechanism of action.

## 2. Results

### 2.1. Chemistry

The synthetic procedure for obtaining the new Met-ITC hybrid is described in Figure 1. Met is obtained by treating Met hydrochloride with a 1 M NaOH solution. The water is evaporated, and the residue is suspended in methanol; the suspension is filtered and finally evaporated to yield basic Met. Met reacts with CS_2_ in the presence of NEt_3_ and *p*-toluenesulfonyl chloride at 0 °C. After acidification with 1 N HCl, the desired product is extracted using Et_2_O and is finally purified via flash chromatography.

### 2.2. Amperometric Recording of H_2_S Release

In the cell-free amperometric assay, Met-ITC exhibited thiol-dependent H_2_S release. Incubation of Met-ITC 1 mM in the presence of L-Cysteine 4 mM resulted in a rapid and sustained generation of H_2_S followed by a gradual decrease (Figure 1). After about 6 min of incubation, Met-ITC showed the maximum concentration of released H_2_S (Cmax~8 µM) and, after 30 min of recording, H_2_S levels reached a stable steady state of approximately 2 µM. In the absence of L-Cysteine, Met-ITC did not release H_2_S. This behavior can be attributed to the reactivity of isothiocyanate moiety with organic thiols. As demonstrated by Lin and colleagues, the N=C=S moiety reacts with L-Cysteine through nucleophilic substitution, leading to the generation of an unstable dithiocarbammate derivative that spontaneously undergoes intramolecular cyclization and releases H_2_S [41]. The ability of ITCs to release H_2_S has also been widely reported in previous experiments [42].

### 2.3. Evaluation of the Release of Met from Met-ITC

The ability of Met-ITC to act as a H_2_S donor prodrug of Met, i.e., the capacity to simultaneously release H_2_S and Met, was evaluated using HPLC analysis. A sample containing Met-ITC (1), dissolved in phosphate buffer (pH = 7.4) at a concentration of 1 mM in the presence of 4 mM L-Cysteine, was diluted in methanol up to a concentration of 0.05 mM (SOLUTION A) and analyzed through HPLC (see details in the experimental section) 30 min after the preparation of the sample itself. The chromatogram (Figure 2A) revealed the presence of two peaks: one relating to Met-ITC (1) (retention time = 2.73 min, 93.5%) and one with a retention time of 11.0 min (6.5%). The analysis was then repeated on a sample enriched with Met (2), obtained by mixing equal volumes of solution A and a solution containing 1 mg/mL of Met (2) in methanol: the chromatogram (Figure 2B) still showed the same two peaks and an increase in the area of the peak with a retention time of 11.0 min, which has therefore been attributed to Met (2).

### 2.4. Effects on Cell Viability of AsPC-1 and MIA PaCa-2 Pancreatic Cancer Cells

Two different pancreatic cancer cell lines were used in this work: AsPC-1 and MIA PaCa-2. The concentrations of Met and Met-ITC ranged from a non-cytotoxic effect, established by 100% cell viability compared to vehicle, to an almost complete abolishment of cell viability (>10% versus vehicle). In AsPC-1 cells, the incubation of increasing concentrations of Met for 72 h did not affect cell viability (Figure 3a). In contrast, the treatment of AsPC-1 cells with Met-ITC for 72 h (Figure 3b) led to a concentration-dependent decrease in cell viability, with an IC50 value of 234.5 µM (Table 1). Of note, the highest concentration of Met-ITC, i.e., 1 mM, evoked an almost total reduction in cell viability (% of cell viability vs. vehicle: 6.1 ± 3.3), while the same concentration of Met did not show a significant inhibitory effect (% of cell viability vs. vehicle: 94.8 ± 3.5).

In MIA PaCa-2 cells, the incubation of Met for 72 h (Figure 4a) resulted in a significant and concentration-dependent decrease in cell viability, with an IC50 value of 7.9 mM (Table 1). However, the highest concentration of Met (10 mM) did not evoke a total reduction in cell viability (% of cell viability vs. vehicle: 34.5 ± 2.0 for Met 10 mM). In contrast, the treatment of MIA PaCa-2 cells with Met-ITC for 72 h (Figure 4b) led to an almost total reduction in cell viability even at concentrations below 10 mM (% of cell viability vs. vehicle: 10.0 ± 5.2 for Met-ITC 1 mM, 8.1 ± 3.4 for Met-ITC 5 mM and 4.0 ± 0.5 for Met-ITC 10 mM). The IC50 value is 370.8 µM (Table 1), suggesting that the hybrid Met-ITC is more potent and more effective than Met in inhibiting MIA PaCa-2 cell viability.

### 2.5. Effects on Cell Viability of MCF-7 Breast Cancer Cells

The antiproliferative effects of Met and Met-ITC were evaluated on MCF-7 breast cancer cells. The concentrations of Met and Met-ITC range from a non-cytotoxic effect, established by 100% cell viability compared with the vehicle, to an almost complete abolishment of cell viability (>10% versus vehicle). As shown in Figure 5a, the incubation of MCF-7 cells with Met for 72 h led to a significant decrease in cell viability only at the highest concentration tested (% cell viability vs. vehicle: 44.0 ± 4.4 for Met 5 mM), with an IC50 value of 5.5 mM (Table 1). In contrast, the treatment of MCF-7 cells with Met-ITC for 72 h (Figure 5b) evoked a concentration-dependent reduction in cell viability. Notably, Met-ITC induced an almost total reduction in cell viability even at concentrations lower than 5 mM (% of cell viability vs. vehicle: 1.3 ± 0.6 for Met-ITC 500 µM, 2.9 ± 1.8 for Met-ITC 1 mM and 1.1 ± 0.3 for Met-ITC 5 mM). The IC50 value is 18 µM (Table 1), indicating that the hybrid compound is more potent and more effective than Met in inhibiting the proliferation of the MCF-7 cell.

### 2.6. Effects on Cell Viability of Non-Tumorigenic Breast Epithelial Cells (MCF-10A)

To exclude potential toxicity in non-cancer cells, we evaluated the cytotoxic effects of Met and Met-ITC in breast epithelial cells (MCF-10A). The incubation of Met for 72 h (Figure 6a) significantly reduced MCF-10A cell viability at the highest concentrations tested, namely 1 and 5 mM, with cell viabilities of 69.3 ± 1.7% and 33.8 ± 4.3%, respectively, and an IC50 value of 2.2 mM (Table 1). The treatment of MCF-10A cells with Met-ITC (Figure 6b) evoked a concentration-dependent decrease in cell viability, with an IC50 value of 171.3 µM (Table 1). These results suggest a slight selectivity in the antiproliferative effect of Met-ITC toward MCF-7.

### 2.7. Intracellular Release of H_2_S in AsPC-1 and MIA PaCa-2 Pancreatic Cancer Cells

The fluorometric recording of H_2_S release demonstrated that Met-ITC can cross the cell membrane and generate H_2_S in a concentration-dependent manner within pancreatic cancer cells. Indeed, incubation of increasing concentrations of Met-ITC (100 µM, 500 µM, 1 mM) in AsPC-1 (Figure 7a,b) and MIA PaCa-2 (Figure 7c,d) cells resulted in a slow and sustained increase in fluorescence emitted by the WSP-1 probe, which is proportional to the amount of H_2_S released. The tested concentrations were chosen based on the cell viability results, selecting 1 mM as the concentration evoking approximately 90% of cell growth inhibition, 500 µM evoking about 50% of the antiproliferative effect and 100 µM evoking an antiproliferative effect < 30%. The generation of H_2_S evoked by the highest concentration of Met-ITC (1 mM) within AsPC-1 cells, although not statistically different, results in a fluorescence increase higher than that produced by the reference compound diallyl disulfide (DADS, 100 µM); at the same time, the generation of H_2_S evoked by Met-ITC 500 µM was almost superimposable with that exhibited by DADS 100 µM. In MIA PaCa-2 cells, the amount of H_2_S released by Met-ITC (1 mM) was about 30% of that shown by DADS (100 µM).

### 2.8. Intracellular Release of H_2_S in MCF-7 Breast Cancer Cells

Figure 8a,b show the H_2_S release recorded in MCF-7 breast cancer cells after incubation of Met-ITC (10 µM, 100 µM and 1 mM). In this cell line, only the highest concentration (1 mM) led to a significant increase in FI, which is proportional to the levels of H_2_S generated within MCF-7 cells. The tested concentrations were chosen on the basis of the cell viability results, selecting 1 mM as the concentration evoking approximately 90% of cell growth inhibition, 100 µM evoking about 50% of antiproliferative effect and 10 µM evoking an antiproliferative effect < 30%. The amount of H_2_S released by Met-ITC (1 mM) was the only one with a similar response to that recorded after the incubation of the reference H_2_S donor DADS (100 µM) (Figure 8a,b).

This experimental procedure revealed the ability of Met-ITC to cross the cell membrane and react with cytosolic thiols, leading to the generation of appreciable amounts of H_2_S during 50 min of treatment when incubated at 500 µM and 1 mM, while Met-ITC 100 µM did not promote any fluorescence increase. Looking at the cell viability results, Met-ITC 100 µM evoked a significant antiproliferative effect when incubated for 72 h in AsPC-1 and MCF-7 cell lines (Section 2.4 and Section 2.5). The different duration of cell viability evaluation (72 h) and H_2_S measurements (50 min) justifies the lack of increase in fluorescence when Met-ITC is incubated at 100 µM: the WSP-1 approach cannot be pushed for longer periods of time because of the possible decay of WSP-1 fluorescence, as reported in the standard operative procedure [43], and relatively high concentrations of Met-ITC must be used to obtain a qualitative result.

### 2.9. Intracellular Release of H_2_S in MCF-10A Non-Tumorigenic Breast Epithelial Cells

In non-tumorigenic breast cells (MCF-10A), neither the reference H_2_S donor DADS (100 µM) nor the hybrid molecule Met-ITC (10 µM, 100 µM and 1 mM) released H_2_S (Figure 9a,b). Indeed, no increase in FI was detected within this cell line after the incubation of both compounds with the WSP-1 dye.

### 2.10. Inhibition of Cell Cycle Progression in MCF-7 Breast Cancer Cells

In preliminary experiments aimed at evaluating the effects of Met-ITC on the cell viability of cancer cells (Section 2.4 and Section 2.5), MCF-7 was identified as the most sensitive cell line (Table 1). Therefore, we selected breast cancer cells to study the potential mechanisms of action of Met-ITC. First, we evaluated the effects of Met-ITC on cell cycle progression. We performed this set of experiments using Met-ITC concentrations close to 20 µM, since the calculated IC50 value for the antiproliferative effects of Met-ITC was approximately 20 µM. The incubation of Met-ITC (10, 20 and 100 µM) for 72 h led to a concentration-dependent change in the percentage of MCF-7 cells in each phase of the cell cycle. Specifically, the treatment of MCF-7 breast cancer cells with Met-ITC reduced the number of cells in the G0/G1 phase and increased the number of cells in the S and G2/M phases (Figure 10 and Table 2).

### 2.11. Pro-Apoptotic Effects of Met-ITC in MCF-7 Breast Cancer Cells

The effects of Met-ITC on two markers of apoptosis were then investigated. First, we focused on early apoptosis by measuring annexin V expression in MCF-7 cells. After 72 h of treatment, Met-ITC (10, 20 and 100 µM) significantly increased the number of cells in the early phase of apoptosis (% of apoptotic cells: 27.1 ± 1.9 for vehicle, 36.2 ± 1.9 for 10 µM Met-ITC, 38.0 ± 0.7 for 20 µM Met-ITC and 42.2 ± 0.9 for 100 µM Met-ITC) (Figure 11a,b). Then, the effects of Met-ITC on markers of mild/late apoptosis (i.e., caspase 3/7) were evaluated. As shown in Figure 11c,d, the incubation of Met-ITC (10, 20 and 100 µM) for 72 h increased the number of cells in the mild/late phase of apoptosis in a concentration-dependent manner (% apoptotic cells: 29.5 ± 2.3 for vehicle, 29.4 ± 2.7 for 10 µM Met-ITC, 50.9 ± 2.6 for 20 µM Met-ITC and 75.5 ± 1.2 for 100 µM Met-ITC).

## 3. Discussion

The interest in the potential repurposing of Met in cancer started in 2005 after the publication of an observational study in which Evans and colleagues reported pharmacoepidemiologic evidence of a reduced risk of cancer in diabetic patients treated with Met [44]. Since that time, several pre-clinical and clinical studies have been designed to confirm this observation and explore the potential mechanism of action of Met. Among them, some observational clinical trials suffered time-related bias [45]; on the other hand, in pre-clinical studies, Met has confirmed its efficacy as anti-cancer drug, but as already observed for its anti-diabetic effect, a clear mechanism of action has not been identified. Indeed, many mechanisms of action have been proposed for its anti-cancer activity, such as the inhibition of the mitochondrial respiratory chain complex and the consequent activation of the AMPK pathway or the lowering of insulin and insulin-like growth factor 1 (IGF-1) levels, which reduce the activation of the PI3K/Akt/mTOR pathway. More generally, Met demonstrated the ability to evoke anti-cancer effects in several pre-clinical tumor models or cell lines via different pathways, all resulting in the inhibition of cancer cell proliferation [46]. Met has been reported to arrest the cell cycle in the G1 phase of several breast cancer cell lines (including MCF-7) when incubated at mM concentration levels (about 2 mM). The effect has been reported to be concentration-dependent [7]. The effect of H_2_S and H_2_S donors in cancer has been widely reported in several experimental studies and, similarly to what observed for Met, it seems to act via several mechanisms ranging from the alteration of the cell cycle and the DNA replication pathway [47] to apoptotic cell death [48] induced via the inhibition of ERK1/2 phosphorylation [20], the attenuation of ferroptosis [49] or autophagy [19]. Among the different H_2_S donors, isothiocyanates are well known for their peculiar epigenetic anti-cancer effect, linked to their ability to inhibit histone deacetylases (HDACs) enzymes [50,51]. Based on these findings, we designed a hybrid molecule, Met-ITC, able to join the well-known anti-diabetic drug Met with a H_2_S donor isothiocyanate moiety, which is potentially useful as a new anti-cancer agent.

Indeed, the results of this study demonstrated that the hybrid molecule, Met-ITC, was able to release H_2_S (thanks to the isothiocyanate moiety) and was also able to release native Met. Moreover, Met-ITC was also designed to overcome certain unfavorable physicochemical features of Met, which limit its pharmacokinetic profile. For instance, the high levels of the hydrophilicity of Met limit its passive diffusion and accounts for a relatively low oral bioavailability. In fact, Met-ITC exhibited a LogP of 0.39, which is more favorable than that of Met (−1.83) (the values of LogP are theoretically calculated by means of software ALOGP 2.1). However, the evaluation of pharmacokinetic profiles of Met-ITC is not the aim of this study and will be carried out in future investigations on appropriate experimental models.

In all experiments performed on different cancer cell lines, Met-ITC showed greater efficacy and potency than Met. In pancreatic cell lines, and in particular in AsPC-1, native Met did not affect cell viability while in MIA, PaCa-2 showed modest efficacy and potency in reducing cell viability. On the other hand, Met-ITC showed a similar increased effect on both the two pancreatic cancer cell lines and was even more potent on AsPC-1. This effect could be associated with the different behavior exhibited by Met on different pancreatic cancer cell lines depending on the environment. In particular, Zechner et al. observed that, depending on the glucose concentrations in the medium, Met could induce the apoptosis of pancreatic cancer cells or inhibit gemcitabine-induced apoptosis in these cell lines [52]. This kind of behavior may justify the variable effect of Met on AsPC-1 and MIA PaCa-2, depending on their different glucose consumption during Met treatment. Anyway, despite the different effect induced by native Met, the enhancement in the anti-proliferative effect observed with the administration of Met-ITC in all the cancer cell lines seems to confirm that the improvement in the anti-proliferative effect is due to the presence of the isothiocyanate moiety and to the molecule’s ability to release H_2_S.

Among the tested cancer cell lines, the breast cancer cell line MCF-7 was the most sensitive to the new hybrid molecule Met-ITC. A further investigation on breast cells allowed us to observe that the anti-proliferative effect of Met-ITC (but not of Met) on a non-tumorigenic cell-line (MCF-10A) was significantly lower than that observed in MCF-7. At the same time, MCF-10A were the only cells in which the increase in intracellular H_2_S was not observed after Met-ITC administration. This evidence suggested that the additive anti-proliferative effect showed by Met-ITC was likely to be related to the presence of the H_2_S donor moiety and supported some selectivity of Met-ITC against tumor cell lines.

Finally, in the most sensitive cell line, MCF-7, the hybrid compound Met-ITC was confirmed to be endowed with mechanisms of action described in the literature for the anti-cancer effect of both Met and H_2_S donors. Indeed, it affects the cell cycle and promotes both early and late apoptosis in a concentration-dependent manner.

## 4. Materials and Methods

### 4.1. Chemistry

All chemicals were purchased from Merck Life Science (Milan, Italy) and were of the highest purity. Reactions were routinely monitored via TLC performed on aluminum-backed silica gel plates (Merck DC, Alufolien Kieselgel 60 F254) with spots visualized by UV light (λ = 254, 365 nm). Solvents were removed using a rotary evaporator operating at a reduced pressure of ~10 Torr. Organic solutions were dried over anhydrous Na_2_SO_4_. Chromatographic separations were performed on silica gel (silica gel 60, 0.015−0.040 mm; Merck DC) columns. The uncorrected melting points were determined using a Reichert Köfler hot-stage apparatus. ^1^H and ^13^C NMR spectra were recorded at 400 and 100 MHz, respectively, on a Bruker AVANCE 400. The coupling constants are given in Hertz. Chemical shifts (δ) were reported in parts per million (ppm) relative to the internal reference tetramethylsilane (TMS). High resolution mass spectrum was recorded on a Thermo LTQ Orbitrap XL in electrospray positive ionization modes (ESI-MS). The purity of the Met-ITC was determined, using a Shimadzu LC-20AD SP liquid chromatograph equipped with a DDA Detector (λ = 254 nm) with a C18 column (250 mm × 4.6 mm, 5 μm, Shim-pack). The mobile phase, delivered at isocratic flow, consisted of acetonitrile (50%) and water (50%) and a flow rate of 1.0 mL/min. Met-ITC showed a purity value percentage of ≥95%. Carbon, hydrogen, nitrogen and sulphur analyses were performed on a Vario MICRO cu-be instrument (Elementar) and agreed with the theoretical values to within (0.4%).

Met-ITC and *N*^1^,*N*^1^-dimethyl-*N*^4^-isothiocyanatobiguanidine. A solution of Met hydrochloride (*N*^1^,*N*^1^-dimethylimidodicarbonimidicdiamide hydrochloride) (1.0 g, 6.0 mmol) in 10 mL of 1 M NaOH was stirred for 30 min at room temperature. Water was then evaporated under reduced pressure and the residue was suspended in 30 mL of MeOH. The solvent was evaporated, and the residue was taken up with 20 mL of MeOH. NaCl is removed by filtration under vacuo and the filtrate was evaporated to yield basic Met 2 (0.772 g, yield = 99%). To a solution of Met (0.160 g, 1.20 mmol) in 1.0 mL of dry THF, NEt_3_ (0.75 mL, 5.4 mmol) and CS_2_ (0.16 mL, 2.6 mmol) were added dropwise at 0 °C. The mixture was stirred at 0 °C for 15 min and then left at room temperature for one hour. Then, *p*-toluenesulfonyl chloride (0.297 g, 1.56 mmol) in 1.0 mL of the same solvent was added at 0 °C and the obtained mixture was stirred at room temperature for one hour. Subsequently, 1.5 mL of 1N HCl and Et_2_O (2.0 mL) were added, and the reaction mixture was stirred for 5 min at room temperature and extracted with Et_2_O. The organic phase was dried over Na_2_SO_4_, filtered and finally evaporated under reduced pressure. The crude product obtained was purified via flash chromatography on silica gel (AcOEt: petroleum ether 40–60 °C = 7:3 as the eluant), providing the desired derivative (Met-ITC). Mp = 225 °C. ^1^H-NMR (400 Hz, DMSO-*d*_6_): δ 6.97 (3H, bs), 3.01 (6H, s). ^13^C-NMR (100 Hz, DMSO-*d*6): δ 176.53, 166.02, 164.36, 36.01. HRMS (ESI) *m*/*z* Calcd for C_5_H_10_N_5_S^+^: 172.06514; Found: 172.06525 [M+H]^+^. Anal. Calcd for C5H9N5S: C, 35.07; H, 5.30; N, 40.90; S, 18.72. Found: C, 34.96 H, 4.97; N, 40.63; S, 19.05.

#### HPLC Analysis of Met and Met-ITC

The release of Met (2) from Met-ITC (1) was performed on the HPLC system using a Shimadzu LC-20AD SP apparatus with a DDA detector at 254 nm (C18 column (250 mm × 4.6 mm, 5 µm, Shim-pack). The mobile phase, delivered in isocratic flow, consisted of acetonitrile and a 10 mM ammonium acetate buffer solution (pH 5.0) at the ratio of 90:10 (*v*/*v*) at a flow rate of 1.0 mL/min at room temperature.

### 4.2. Amperometric Measurement of H_2_S Release

The H_2_S-releasing properties of Met-ITC were evaluated using H_2_S-selective minielectrodes connected to the Apollo-4000 Free Radical Analyzer (World Precision Instruments-WPI, FL, USA) and immersed in 2 mL PBS buffer (Merck KGaA, Darmstadt, Germany) at room temperature and pH 7.4 as fully described previously [53]. Once a stable baseline was reached, Met-ITC dissolved in DMSO (Merck KGaA, Darmstadt, Germany) was incubated in the PBS buffer to a final concentration of 1 mM. The release of H_2_S was monitored for 30 min. When required by the experimental protocol, 4 mM L-Cysteine (Merck KGaA, Darmstadt, Germany) was dissolved in PBS buffer to mimic the endogenous presence of organic thiols. The H_2_S concentration generated after the incubation of Met-ITC was determined using NaHS 1 µM as a reference H_2_S donor (Cayman Chemical, Ann Arbor, Michigan, USA) incubated in PBS buffer, pH 4.0. Furthermore, the correct relationship between the amperometric currents (recorded in pA) and the corresponding concentrations of H_2_S was previously determined by suitable calibration curves, which were obtained using NaHS (1–3–10 μM) incubated in PBS buffer at pH 4.0.

### 4.3. Cell Cultures

Cells were cultured in tissue flasks (T-75) at 37 °C in a CO_2_ (5%) incubator. The appropriate culture medium was used for each cell line as described below. All supplements were purchased from Merck KGaA, Darmstadt, Germany.

#### 4.3.1. Human Pancreatic Cancer Cells

Human pancreas adenocarcinoma ascites metastasis cells (AsPC-1; Merck KGaA, Darmstadt, Germany) were cultured in basal medium (RPMI-1640; Merck KGaA, Darmstadt, Germany) supplemented with 10% fetal bovine serum, 2 mM L-glutamine, 1 mM sodium pyruvate and 100 mg/mL streptomycin plus 100 U/mL penicillin. Human pancreatic carcinoma cells (MIA PaCa-2; Merck KGaA, Darmstadt, Germany) were cultured in basal medium (DMEM-high glucose; Merck KGaA, Darmstadt, Germany) supplemented with 10% fetal bovine serum, 2 mM L-glutamine and 100 mg/mL streptomycin plus 100 U/mL penicillin.

#### 4.3.2. Human Breast Cancer Cells

Human breast adenocarcinoma cells (MCF-7; Merck KGaA, Darmstadt, Germany) were cultured in basal medium (EMEM; Merck KGaA, Darmstadt, Germany) supplemented with 10% fetal bovine serum, 2 mM L-glutamine, 1% non-essential amino acids and 100 mg/mL streptomycin plus 100 U/mL penicillin.

#### 4.3.3. Nontumorigenic Breast Epithelial Cells

The non-malignant breast epithelial cell line MCF-10A (Merck KGaA, Darmstadt, Germany) was cultured in a Mammary Epithelial Cell Growth Medium Bullet Kit (MEGM; Merck KGaA, Darmstadt, Germany) supplemented with cholera toxin (100 ng/mL; Merck KGaA, Darmstadt, Germany).

### 4.4. Cell Viability

Cells were grown to 90% confluence and seeded in clear 96-well plates at a density of 10^4^ cells/well (3 × 10^3^ cells/well for MCF-10A). After 24 h, the culture medium was replaced with fresh medium and the cells were treated with vehicle (DMSO 1%), Met (100 µM–10 mM) or Met-ITC (1 µM–10 mM) for 72 h depending on the cell line. The concentrations of Met and Met-ITC range from a non-cytotoxic effect, established by 100% cell viability compared with the vehicle, to an almost complete abolishment of cell viability (>10% versus vehicle). At the end of treatment, water-soluble tetrazolium salt-1 (WST-1; Roche, Basel, Switzerland) was added to each well (1:10) and the plate was placed for 1 h in a CO_2_ (5%) incubator at 37 °C. Cell viability was assessed with a microplate reader EnSpire (PerkinElmer, Waltham, MA, USA) at λ = 495 nm.

### 4.5. Intracellular Release of H_2_S

Cells were seeded in 96-well black plates at a density of 72 × 10^3^ cells/well. After 24 h, the culture medium was replaced with 180 µL of a freshly prepared solution of the Washington State Probe-1 (WSP-1) dye (Cayman Chemical, Ann Arbor, MI, USA), dissolved in DMSO obtaining a stock solution of 2 mM and then diluted in buffer standard at the final concentration of 100 µM (buffer composition: KCl 2 mM, NaCl 120 mM, CaCl_2_·2H_2_O 2 mM, HEPES 20 mM, glucose 5 mM and MgCl_2_·6H_2_O 1 mM; pH 7.4). WSP-1 is a highly selective probe for detecting H_2_S and rapidly reacts with the gaseous molecule, releasing a fluorophore [43]. To allow the cells to upload the dye, WSP-1 was incubated at 37 °C in a CO_2_ (5%) incubator for 30 min. Then, the probe in excess (i.e., the probe that did not enter the cells) was removed and 180 µL of buffer standard was added. After the assessment of baseline fluorescence index (FI), vehicle (DMSO 1%), 100 µM diallyl disulfide (DADS) or Met-ITC were added. The concentrations of Met-ITC were chosen based on cell viability results, depending on the cell line. The H_2_S donor DADS, a natural polysulfide derived from Alliaceae family [54], was used as the reference compound. All compounds were dissolved in DMSO and diluted in buffer standard. The increase in fluorescence, corresponding to the release of H_2_S by the tested compounds and expressed as FI, was monitored every 5 min at λex = 465 nm and λem = 515 nm for 50 min with the microplate reader EnSpire (PerkinElmer, Waltham, MA, USA).

### 4.6. Cell Cycle Analysis

MCF-7 cells were selected for this set of experiments because Met-ITC showed the most effective cytotoxic effect. Cells were grown to 90% confluence and seeded in clear 6-well plates at a density of 5 × 10^5^ cells/well. After 24 h, the culture medium was replaced with fresh medium, and the cells were treated with vehicle (DMSO 1%) or Met-ITC (10, 20 and 100 µM) for 72 h. At the end of treatment, the cell cycle was analyzed via flow cytometry with the Muse™ Cell Cycle Kit (Merck KGaA, Darmstadt, Germany). Briefly, cells were harvested with a scraper, washed once with PBS (Merck KGaA, Darmstadt, Germany) and fixed at −20 °C in 70% ice-cold ethanol (1 mL) for a minimum of 3 h. Then, the cells were washed again with PBS, resuspended in Muse™ Cell Cycle Reagent (106 cells/200 µL) and incubated in the dark at 37 °C for 30 min. The distribution of cells in different phases of the cell cycle (G0/G1, S and G2/M) was analyzed on the Muse™ Cell Analyzer (Merck KGaA, Darmstadt, Germany).

### 4.7. Annexin V Detection

MCF-7 cells were seeded in clear 6-well plates at a density of 5 × 10^5^ cells/well. The next day, they were treated as in Section 2.5. At the end of the treatment, Annexin V was detected with the Muse™ Annexin V & Dead Cell Kit (Merck KGaA, Darmstadt, Germany). According to the manufacturer’s protocol, cells were harvested with a scraper, resuspended in culture medium (100 µL) to a final concentration between 10^5^ and 10^7^ cells/mL and incubated with Muse™ Annexin V & Dead Cell Reagent (100 µL) for 20 min at room temperature in the dark. The percentage of early apoptotic cells was analyzed using flow cytometry on the Muse™ Cell Analyzer (Merck KGaA, Darmstadt, Germany).

### 4.8. Caspase 3/7 Activity

MCF-7 cells were seeded in clear 24-well plates at a density of 10^5^ cells/well. After 24 h, they were treated as in Section 2.5. At the end of treatment, caspase 3/7 activity was analyzed with the Muse™ Caspase 3/7 Activity Kit (Merck KGaA, Darmstadt, Germany). According to the manufacturer’s protocol, cells were harvested with a scraper, washed with PBS (Merck KGaA, Darmstadt, Germany) and resuspended in Assay Buffer 1× to a final concentration between 1 × 10^5^ and 5 × 10^6^ cells/mL. Muse™ Caspase-3/7 Reagent working solution was incubated (1:10) at 37 °C for 30 min and then 150 μL of Muse™ Caspase 7-AAD working solution was added in the dark at 37 °C for 5 min. The percentage of cells in the mild stage of apoptosis was analyzed via flow cytometry on the Muse™ Cell Analyzer (Merck KGaA, Darmstadt, Germany).

### 4.9. Data Analysis

At least three separate experiments were performed, each in triplicate. Results are shown as mean ± SEM. Statistical analysis (one-way ANOVA followed by Bonferroni’s post hoc test) was performed using GraphPad Prism software (version 5.0; San Diego, CA, USA). Statistical significance was set at *p*-value < 0.05.

## 5. Conclusions

The concept of developing hybrid molecules represents a strategy to enhance the pharmacological effects of “old” drugs. In this work, a proof of concept is provided concerning the features of Met-ITC as novel hybrid drug, namely the addition of a chemical moiety able to release H_2_S, which is a gasotrasmitter known for a plethora of beneficial effects, including anti-cancer properties. Met-ITC was able to promote a stronger anti-proliferative effect than Met in different cancer cell lines, indicating that the addition of ITC improved anti-cancer activities.

## 6. Patents

International Patent “Metformin conjugates and therapeutic use thereof” WO 2023/017400 A1 international publication date: 16 February 2023. Authors: CALDERONE Vincenzo, MARTELLI Alma, CITI Valentina, PIRAGINE Eugenia, TESTAI Lara, DA SETTIMO PASSETTI Federico, TALIANI Sabrina, BARRESI Elisabetta.

## Data Availability

The data presented in this study are available on request from the corresponding author.

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
