# Peer review of "Anti-Proliferative Properties of the Novel Hybrid Drug Met-ITC, Composed of the Native Drug Metformin with the Addition of an Isothiocyanate H_2_S Donor Moiety, in Different Cancer Cell Lines"

_ijms, 2023, doi:10.3390/ijms242216131_

Round 1
Reviewer 1 Report
Comments and Suggestions for Authors
The herein presented paper, by Citi et al., presents a metformin ITC conjugated drug that has potential antitumor properties in human pancreatic and breast cancer cells and compared the effects to the normal cell line MCF10A.
The synthesis of the Met-ITC is well described and all the information provided is useful for reproducibility of the method.
The antiproliferative effect is visible on pancreatic cell, Met-ITC has a dose-dependent effect on cell proliferation. While Met alone has no inhibitory effect on AsPC-1 cells, but with significant effect on MIA PaCa-2.
Major Observation:
Why the tested compounds have different range for MIA PaCa-2 compared to AsPC-1? The dose for Met is higher, same for MeT-ITC. Please represent in Figure 3 and 4 graph with the same range, or JUSTIFY why you used different ranges of concentrations.
The same observation is for figure 5 and 6.
For a better understanding of the effect you must compare all the cells within the same ranges and same conditions. Please explain.
Add in table 1 also a column with the range of concentrations that you tested (e.g. 10 micrograms – 100 nanograms or how it is). The IC50 calculation is influenced by the range of the treatments and the dose.
For the Cell cycle analysis, how did you prepare the cells for this test? Did you perform a starvation step 16-20 h before starting the treatment to keep the cell in G0/G1 at the beginning?
For the apoptosis / necrosis can you please provide the unstained samples plots? It is important to understand how you choose the negative cells. You can prepare supplemental material for this assay.
The conclusion is missing. Chapter 5.
If you cannot answer the comments, your discussions are not supported by the results. You need more details in the results chapter to fully understand how you performed the experiments
Comments on the Quality of English LanguageMinor editing required - check for spelling.
Author Response
Please, see the attached pdf file.

Reviewer 2 Report
Comments and Suggestions for Authors
The synthesis of new molecules with anti-cancer activity is of great interest to scientific society. The submitted manuscript deals with a new hybrid molecules derived from Metformin and possessing isothiocyanate group. The synthetic procedure is based on the reaction of metformin with carbon disulfide followed by the addition of tosyl chloride to the reaction mixture. This synthesis seems to be easy and straightforward, although some basic experimental data should be clarified to the readers. The analysis of structure was based on NMR and HR-MS data. The accuracy of 13C NMR is one digit after coma, not two. The chemical shift of carbon atom from NCS group should appear at around 145 ppm. This signal was not observed. In addition, the product is a crystalline material, and its purity and composition was not evaluated by elemental analysis. The HPLC trace shows, in Figure 2, two peaks with low symmetry. At least two different signals are clearly visible at 2.7 min. This trace is not consistent with single molecule with purity exceeding 95%.
The presence of the second peak at 10.9, on the HPLC trace can be attributed to Metformin. It is highly desirable to use additional methods, like HPLC-MS, to validate this precisely.
Figure 1 shows the fast release of hydrogen sulfite that occurs within 5 minutes. All biological experiments took a much longer time. The question of how it influences the pro-apoptotic effects observed remains open.
Comments on the Quality of English Language
Some minor modification are required.
Author Response
Please, see the attached pdf file.

Round 2
Reviewer 1 Report
Comments and Suggestions for Authors
Dear authors,
Thank you for answering to all my remarks.
I will answer point-by-point:
1. No, you don't need to change the graphs, your argument is sufficient for me and I fully understood the rational of you evaluation.
2. The table looks good and I am glad that you decided to add the tested ranges, thus other researchers can use this information for their own study.
3. The answer for the cell cycle is strong enough, thus I agree with you.
4. Great answer, thank you for the detailed explanation. I agree with your answer.
5. Even if IJMS does not require a separate conclusion, for your study is suitable, you have a lot of data presented and it is helpful when reading your work.
You did answered to all the remarks, and I am more than happy to say that your paper is acceptable for publication.
Congratulations for your work!